# A Spatial Distribution Empirical Model of Surface Soil Water Content and Soil Workability on an Unplanted Sugarcane Farm Area Using Sentinel-1A Data towards Precision Agriculture Applications

Harry Imantho [1,2], Kudang Boro Seminar [3,*], Wawan Hermawan [3] and Satyanto Krido Saptomo [4]

1 Doctoral Student of Agriculture Engineering Science, Faculty of Agricultural Engineering & Technology, IPB University, Bogor 16680, Indonesia
2 Earth Observatory and Change Section, SEAMEO BIOTROP, Bogor 16134, Indonesia
3 Mechanical and Biosystem Engineering Department, Faculty of Agricultural Engineering & Technology, IPB University, Bogor 16680, Indonesia
4 Civil and Environmental Engineering Department, Faculty of Agricultural Engineering & Technology, IPB University, Bogor 16680, Indonesia
* Correspondence: kseminar@apps.ipb.ac.id

**Abstract:** Obtaining soil water content and soil workability data using remote sensing technology with passive sensors has some limitations due to cloud cover, cloud shadow, haze and smoke. This study proposes a method for computing soil water content and soil workability over large areas, faster and in near real-time based on Sentinel-1A (SAR) data. Sample data collected from sugarcane plantations in the Kediri and Sidoarjo districts in East Java, Indonesia, were used to develop a mathematical model of the proposed method using multi-polynomial regression. The performance indicators of the model (RMSE, MAPE and accuracy) were calculated with the results of RMSE = 0.213 and 0.250, MAPE = 16.39% and 18.79%, and accuracy = 83.6% and 81.2% for the training and testing models, respectively. The distribution of soil water content and soil workability can be computed and visualized using a spatial map. The future contribution of this work is to develop a decision support system for the selection of appropriate machinery for sugarcane field operations based on the principles of precision agriculture.

**Keywords:** precision agriculture; radar; Sentinel-1; soil water content; soil workability

## 1. Introduction

Data and information on soil water content are significant in the cycle of global water and affect the surface energy balance at the earth's surface [1]. Obtaining soil water content over a large area is indispensable in the modeling of hydrology, agriculture and meteorology/climatology and plays a key role in estimating the amount of water lost through evapotranspiration and irrigation needs [2–8]. Hence, spatial and temporal soil water content is critical information for precision agricultural practices. According to the International Society of Precision Agriculture (ISPA), precision agriculture is a combination of strategy, methodology and technology to handle and manage the spatial and temporal variability of agricultural land for improved resource use efficiency, productivity, quality, profitability and sustainability of agricultural production.

The amount of water available in the soil affects the transpiration mechanism along with nutrient uptake by plants; thus, the condition of soil water content is an important factor that must be considered in fertilizer applications [9]. In situ measurements over a large area are difficult due to spatial–temporal variability that is caused by heterogeneity of soil texture, topography, vegetation, climatic conditions and the surrounding environment [10,11]. Although the problem is being addressed by passive remote sensing

techniques, i.e., SMOS and SMAP satellites, their spatial resolution is too low (i.e., 1–36 km) and therefore are inadequate for variable rate fertilizer and irrigation applications that require a 5–10 m resolution [12], crop management, water balance assessment and ecological modelling [13–15].

The temporal and spatial variability of soil water content is very important for farm management over large-scale farm areas such as sugarcane plantations. Previous studies have shown that the spatial and temporal variability of soil physical properties and sugarcane growth/productivity are spatially dependent [16,17]. Unfortunately, direct measurements of soil water content over large farm areas are ineffective and inefficient because they are relatively expensive and time consuming for laboratory sampling analysis to cover a large area [10,11,18–20]. However, temporal and spatial variability over large areas such as plantations cannot be ignored; thus, the adoption of remote sensing technology is one of the best choices for understanding spatial and temporal dynamics over intensive agriculture areas.

Land preparation and tillage using farm machinery is intended to modify soil structure within the root zone to create suitable conditions needed by crops. Optimum soil structure in accordance with crop growth and development can be achieved if the tillage is conducted at the optimum soil workability [21]. Soil workability is an optimal condition of soil for tillage and cultivation, which is influenced by soil texture and soil water content. The condition of soil workability within a certain period can be the basis for route planning and scheduling farm machinery operations, and it enables farmers to handle and manage the spatial variability and diversity of land efficiently [22,23].

Several studies of soil–traction interactions have made attempts to achieve optimum performance and field efficiency [24,25]. The traction performance indicators include the slip, drawbar pull, sinkage, traction coefficient and efficiency, and rolling resistance, which are associated with soil density and workability [26,27]. According to Kisu [28] and Kumar et al. [29], the traction performance is also influenced by texture and soil water content.

A number of studies on using multispectral remote sensing data are limited by environmental factors such as cloud cover, cloud shadows, haze and smoke, especially in the tropical areas [30,31]. Mostly agricultural activities are carried out in the rainy season; therefore, it is hard to acquire cloud-free multispectral imagery for soil water content analysis. One of the solutions to cloud-free data acquisition is the use of a High-Resolution Synthetic Aperture Radar (SAR) for exploring and analyzing soil physical properties. The presence of water in the surface of soil or contained in the soil layer affects the reflected signal received by SAR instrument [32]. The SAR is able to monitor the earth's surface without the constraints of clouds, haze and smoke.

The sensitivity of the SAR backscatter to the dielectric constant of the soil is a solid reason for the use of SAR data to detect soil water content [33]. Aside from being influenced by soil dielectric properties, SAR backscattering is also sensitive to vegetation cover and is affected by surface roughness [34]. A number of SAR backscattering separation methods have been proposed to distinguish soil and vegetation signatures, such as the semiempirical Oh model [35] and the Dubois model [36], and the theoretical integral equation model (IEM) and advanced IEM (AIEM) [37,38] for calculating the soil water content above the bare soil base based on TerraSAR-X data. The calculation of soil water content in vegetated areas generally combines the surface microwave model and the canopy model. The Oh, Dubois and IEM surface radiation transfer models were combined with the water cloud canopy model (WCM) to assess the advantages and disadvantages of combining different models to simulate VV polarized radar backscattering for entire vegetation periods from different wheat fields [39]. The study conducted in [40] combined the Oh model with the Water Cloud Model (WCM) to simulate SAR backscattering, and then it developed a function to calculate the surface soil water content in the Nagqu region of the Tibetan Plateau using Sentinel-1 SAR and MODIS data.

The statistical relationship between backscattering SAR and surface soil water content was approached as a linear equation in [41] to estimate soil water content over wheat fields, as an exponential equation [42] and as a polynomial equation [43], evaluating the semi-empirical backscattering model on the L-band and C-band for soybean canopies with soil water content inversion. According to previous studies [44–47], empirical models are reliable for the analysis of soil properties in heterogeneous locations.

The purpose of this study was to develop an empirical model to calculate and map soil water content and soil workability based on Sentinel-1A (radar), which provides high-spatial-resolution imagery data suitable for precision agriculture applications for tillage in plantation areas.

## 2. Materials and Methods

### 2.1. Study Area

The studied sites are located at sugarcane plantation areas of an Indonesian state-owned company (PTPN) X in Kediri and Sidoarjo, East Java (Figure 1). The Kediri plantation area has an altitude of 200 m above sea level, and the rainfall is about 1925 mm year$^{-1}$, whereas the Sidoarjo plantation area is 7 m above sea level with a rainfall of about 1479 mm year$^{-1}$. The sample points were taken from sugarcane areas that had not been replanted after harvesting at both study sites, with a sampling distance of 100 m (Figure 2), which was taken from each 10 ha block. The total sample points taken from Kediri and Sidoarjo were 120 units and 40 units, respectively. The total sampling area covered 15 blocks, consisting of 11 blocks (3%) and 4 blocks (25%) in the Kediri and Sidoarjo districts, respectively. In situ measurements were conducted from June to October 2019.

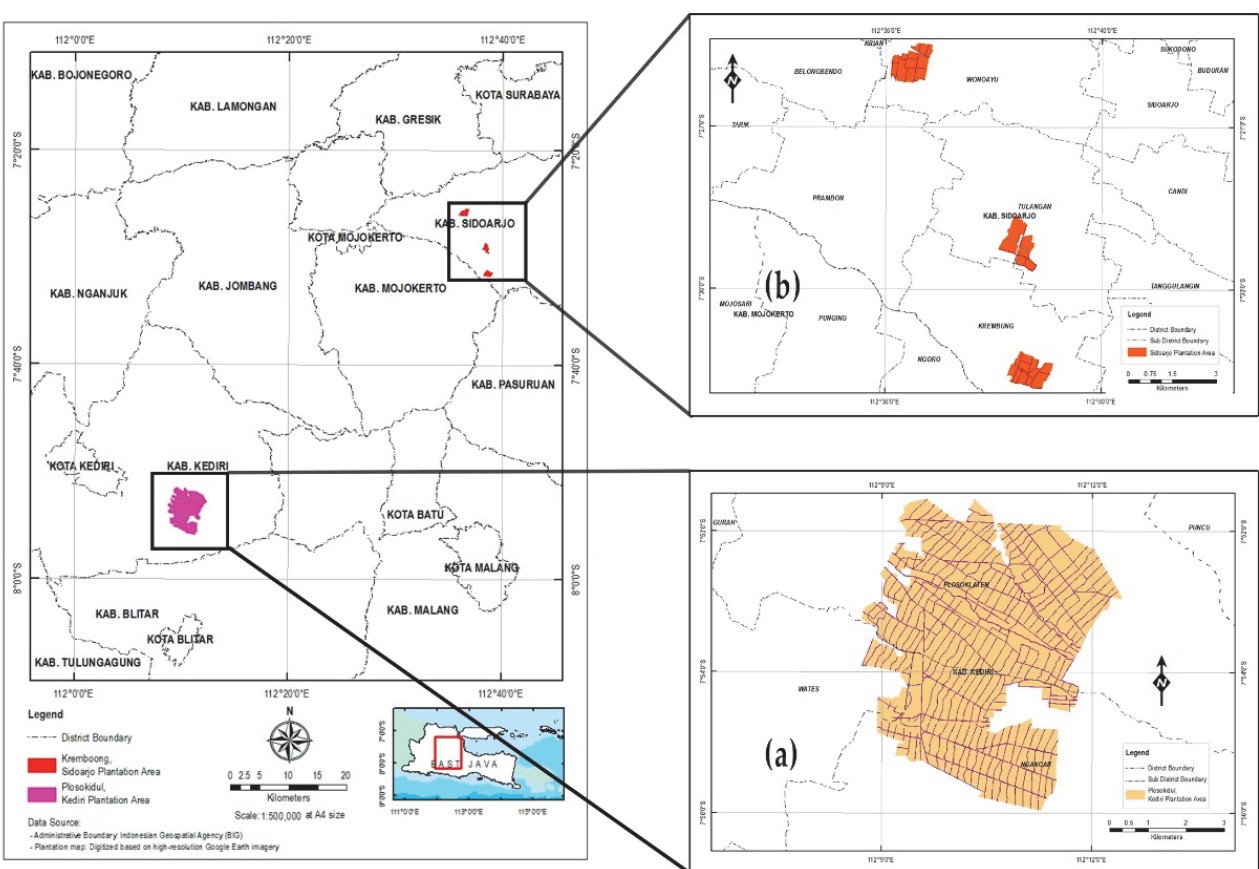

**Figure 1.** Kediri (**a**) and Sidoarjo (**b**) sugarcane plantation area managed by PTPN X.

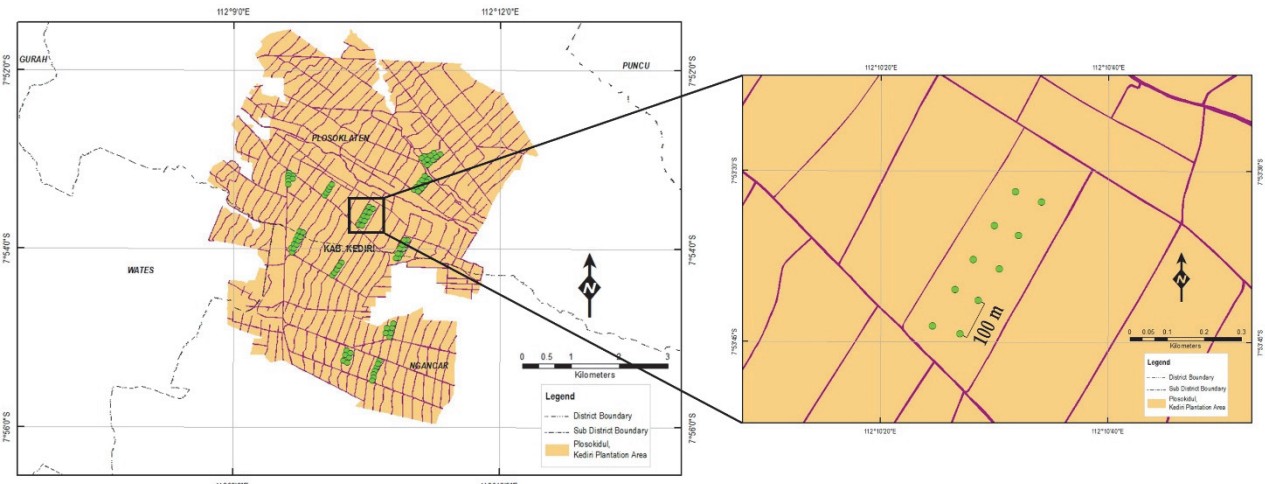

**Figure 2.** Study case area and soil with a sampling distance of 100 m.

## 2.2. Sentinel-1A Data Acquisition

The Sentinel-1 sensor operates on 5.405 GHz of frequency with horizontal and vertical polarization. The Sentinel-1 consists of two polar orbiting satellites, namely Sentinel-1A and Seninel-1B, which were launched by the European Space Agency (ESA) in 2014 and 2016, respectively. Both satellites share an orbital plane with a 180° orbital phasing difference. The Sentinel-1 SAR instrument acquires data in four beam modes, namely Stripmap (SM), Interferometric Wide swath (IW), Extra Wide swath (EW) and Wave (WV). Over land areas, the IW beam mode captures objects using Terrain Observation with the Progressive Scanning SAR (TOPSAR) imaging technique with a coverage area of 250 km in vertical–vertical (VV) co-polarization and vertical–horizontal (VH) cross-polarization. The IW Ground Range Multi-Look Detected High-Resolution (IW GRDH) data have a spatial resolution of 20.4 × 22.5 m, which is open to the public [48]. Sentinel-1A level 1 IW GDRH imageries of both Kediri and Sidoarjo districts were collected from the Sentinel-1 Data Hub website (https://scihub.copernicus.eu/ (accessed on 2 July 2022)). Of the 15 satellite images available from June to October 2019, only 6 images were selected to be processed and analyzed by taking into account the periodicity of the observation period. All selected image data had angles of incidence for ascending and descending overpasses, which were 30.6°–45.9° and 30.4°–46.0°, respectively.

## 2.3. Sentinel-1A Data Processing

The Sentinel Application Platform (SNAP) software was utilized to pre-process the SAR imagery. The process work-flow of the Sentinel-1A IW GRDH product is shown in Figure 3. During the acquisition, the orbit information generated by on-board navigation and geolocation are stored within the Sentinel-1A level 1 data repository. The orbit correction is required to refine the stored orbit position by using precise orbit files provided by the Copernicus Precise Orbit Determination (POD) Service.

The Sentinel-1A level 1 data repository contains some information of thermal noise, border noise and speckle noise, which can decrease the quality of the image. Thermal noise is the additional background energy that interferes with the actual signal, which is below the strength of the thermal energy of the noise [49]. The significant impact of thermal noises on data quality mainly occurs in areas with low mean backscatter, such as calm open waters and rivers. The ESA has included thermal noise information for each image product in the SAFE Sentinel-1 format to support denoising GRD imageries [49]. The border noise is indicated by undesired dark (non-zero) samples at the image boundaries. The border noise causes fluctuations in the background distribution and limits the full exploitation of Sentinel-1 data. The border noise removal module integrated in the Sentinel Application Platform (SNAP) applies a masking approach, sets unwanted dark pixels to zero, and then

removes them by thresholding [50]. The Sentinel-1 Toolbox in the SNAP software provides removal functions for noises, including Mean, Median, Lee, Refined Lee, Lee Sigma, Frost and Gamma-MA to improve the quality of the interpretation and accuracy of backscatter analysis. The Refined Lee filter was applied in this research, since it maintains the details of objects and its boundaries [51–55].

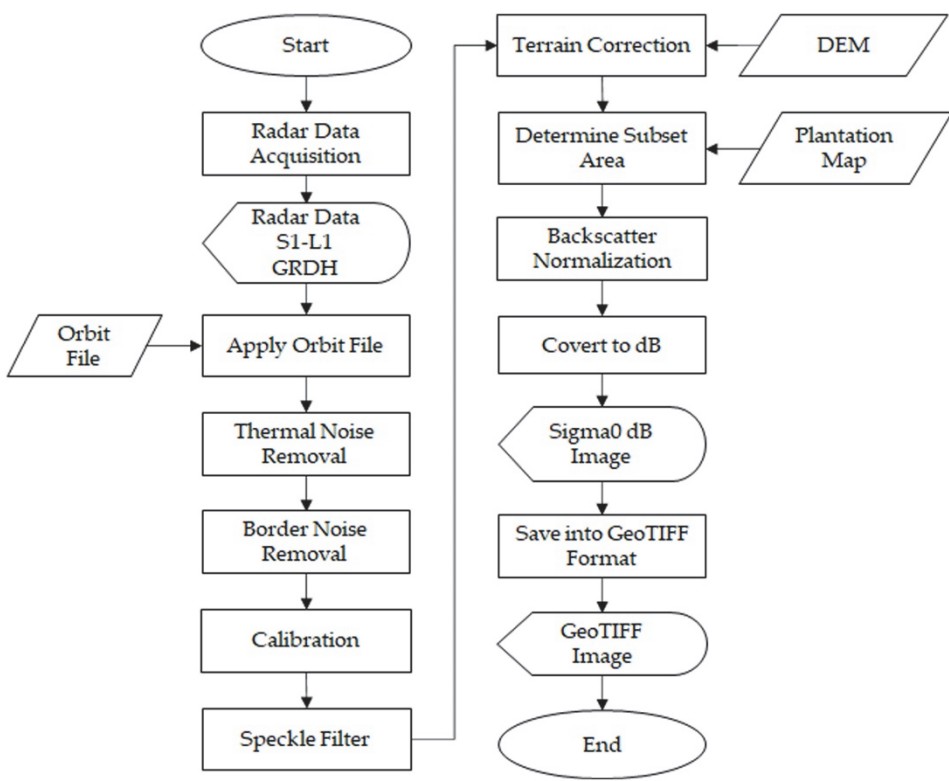

**Figure 3.** The imagery data processing steps (flowchart) using Sentinel-1.

Referring to Figure 3, terrain correction is intended to rectify a scene distortion due to topographical variations and the tilt of the Sentinel-1 wave sensor. Since Sentinel-1A L1 IW GDRH images are produced in zero-Doppler geometry, the Range Doppler Orthorectification method was performed to rectify inherent SAR geometry effects, such as layover, shadows and foreshortening, and re-projecting the data into a ground range coordinate system. The 8 m resolution of the Digital Elevation Model (DEM) provided by Indonesian Geospatial Agency (Badan Informasi Geospasial, BIG) was used on the geometric and terrain correction.

The IW Sentinel-1A beam mode data were captured at viewing angles ranging from 29.1° to 46.0°, causing variations in local angles of incidence at the Kediri and Sidoarjo plantation study sites. Therefore, a backscatter correction was needed using a cosine normalization method [32,56,57] with the following formula:

$$\sigma^{o}_{norm_i} = \frac{\sigma^{o}_{\theta_i} cos^n \left(\theta_{ref}\right)}{cos^n \left(\theta_i\right)} \qquad (1)$$

where $\sigma^{o}_{norm_i}$ is the normalized backscatter value (m$^2$m$^{-2}$), $\theta_i$ is the local incidence angle (degree), $\sigma^{o}_{\theta_i}$ is the backscatter value (m$^2$m$^{-2}$), $\theta_{ref}$ is the reference angle ranging from 30 to 32.9 degrees [48,58] and n is the surface roughness index ranging from 0.24 for forests to 3.36 for savanna [48,59,60].

The observed n value equal to 2 was chosen for the C-band SAR based on previous studies for observing agricultural land surfaces by Lievens et al. [59], van der Velde et al. [60] and Martínez-Agirre et al. [61]. This is also in line with the assumption that ground surface

radiation follows Lambert's cosine law [60]. Afterwards, the normalized backscatter values were transformed using a logarithmic transformation into values of dB units, as follows:

$$\sigma^o(dB) = 10 \cdot \log_{10} \sigma^o_{norm_i} \tag{2}$$

where $\sigma^o$ is the backscatter value (dB), and $\sigma^o_{norm_i}$ is the normalized backscatter (m$^2$m$^{-2}$).

### 2.4. In Situ Soil Sampling and Measurements

At both study sites, soil samples of 15 cm depth were taken within ±2 h when Sentinel-1A passed over the sites. The point coordinates for each soil sample were recorded by a GNSS receiver (GPS). The soil matric suction and soil penetration resistance were measured using a DM8 Tensionmeter at a depth of 15 cm and an Eijkelkamp Penetrograph at a depth of 0–25 cm, respectively, as shown in Figure 4. Soil samples were all taken from study sites using ring samples and were analyzed using a gravimetric method to calculate the soil water content.

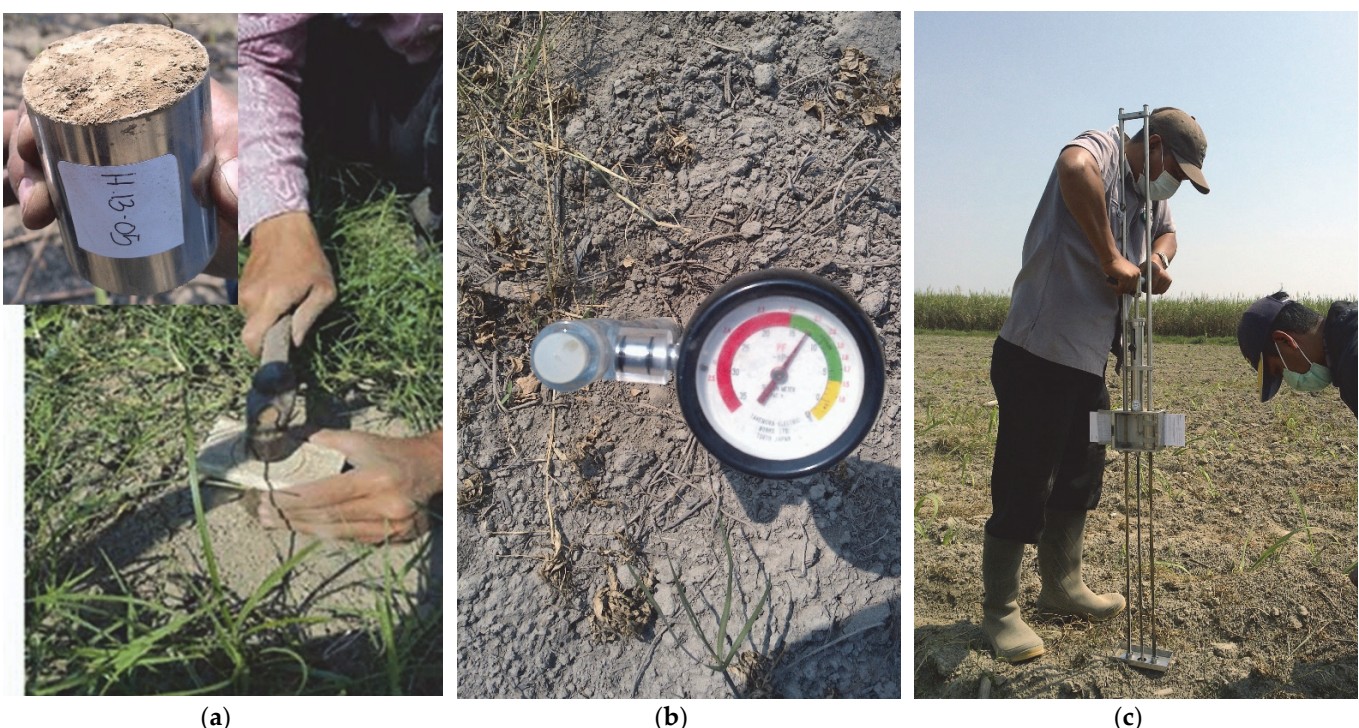

(**a**)                          (**b**)                          (**c**)

**Figure 4.** Soil sampling preparation and measurement: (**a**) Taking the sample for soil water content lab-analysis; (**b**) Soil matric suction measurement; (**c**) soil penetration resistance measurement.

A total of 160 undisturbed soil samples were collected, consisting of 120 samples from the Kediri plantation and 40 samples from the Sidoarjo plantation. Soil composite samples (disturbed soil sample) of each plot were taken to retrieve the soil texture (fraction of soil), soil plasticity index, soil C-organic matter content and soil water content at a field capacity of pF 2.4 and a wilting point of pF 4.2. Moreover, dry bulk density was analyzed to convert the soil water content of a percent of weight (%w) to a percent of volume (%v). The standard deviation, root-mean-square (RMS) error and mean absolute percentage error (MAPE) were used to test the consistency of the measured soil water content and model. Soil penetration resistance measurements were needed to calculate the specific resistance of the plow, which is critical to tillage processes using agricultural machinery that minimizes soil degradation to support precision agriculture site management [62].

*2.5. Spatial Distribution of Soil Water Content and Workability Retrieval Methods*

Referring to the research objectives, to measure the spatial soil water content and soil workability distributions in unplanted sugarcane plantation areas, this was achieved using a multi-polynomial regression method, which correlates volumetric soil water content (%v) and the backscatter of SAR data at the same pixel coordinate positions.

Soil workability is a condition of agricultural land that supports optimal tillage activities without significant obstacles and/or does not have a negative impact on soil structure (such as soil compaction). The calculation of soil workability can be performed using Kretschmer's pedotransfer function [63]. The laboratory plasticity level analyses were done by using Atterberg's method, in accordance with SNI 1966:2008. The optimum soil water content that provides the optimum soil workability was calculated as follows:

$$\theta_{opt} = LPL - 0.15 \times (UPL - LPL) \tag{3}$$

where $\theta_{opt}$ is the soil water content (%vol) which results in optimum soil workability, *LPL* is the lower plasticity level (plastic limit) and *UPL* is the upper plasticity level (liquid limit).

## 3. Results and Discussion

### 3.1. Soil Characteristics of Study Sites

Based on soil laboratory analyses, the Kediri plantation area is classified into loamy sand (USDA soil classification) with the following composition: 75.10% sand, 6.98% loam and 17.10% silt, with C-organic content of about 1.2% m, whereas the Sidoarjo Plantation is formed by 29.5% sand, 22.2% loam and 48.3% silt, classified into silty clay, with 0.8% m C-organic content. The dominance of the sand fraction in the soil in the Kediri plantation causes the Kediri plantation area to be always drier than the Sidoarjo Plantation throughout the season. The dominance of this sand fraction also causes the soil's ability to hold water to be low. This is indicated by the higher porosity values of the Kediri plantation compared to the Sidoarjo plantation, which were 52.18% and 42.98%, respectively. As the porosity of the soil becomes greater, the water loss becomes faster. This ability to measure porosity allows for better irrigation management and for the selection of appropriate agricultural machinery for tillage that conforms to the principles and objectives of precision agriculture.

It is shown in this research that the relationship between soil water content and soil matric suction is specific for soil type, texture and organic matter in different areas. These results are in line with the previous study [64] that discusses the relationship between soil water content and soil matric suction in different specific sites. Figure 5 presents the soil water retention curves (SWRC) of both study sites of our research.

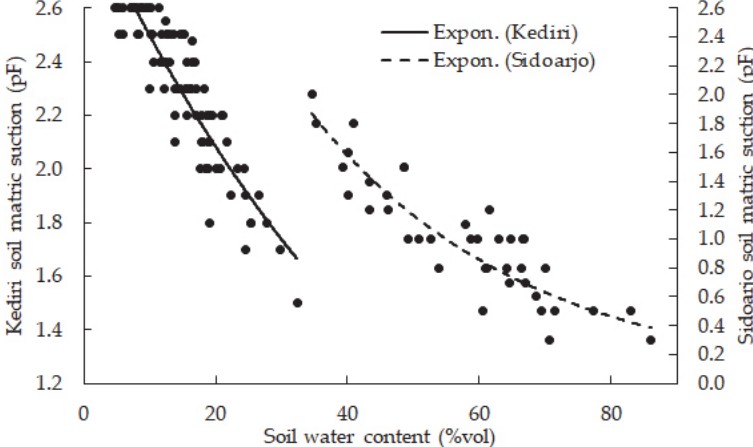

**Figure 5.** Soil water retention curve (SWRC) based on field observations and measurement in Kediri and Sidoarjo plantations.

The field capacity and permanent wilting point of the Kediri plantation was 28.68 ± 6.22 %v and 18.03 ± 3.94 %v, and the Sidoarjo plantation was 29.17 ± 2.78 %v and 20.21 ± 2.57 %v. The Kediri and Sidoarjo plantations had average bulk densities of 1.38 ± 0.095 gr cm$^3$ and 1.41 ± 0.150 gr cm$^3$, respectively.

### 3.2. Developing an Empirical Model of Soil Water Content

The dynamics of soil water content are a function of time, weather and soil/landscape properties. Based on precision agriculture principles, agriculture activities must account for the variability/heterogeneity of landscape conditions; therefore, every action and treatment should have optimum positive impacts based on the discovered variability.

To calculate the spatial distribution of soil water content based on backscatter values, the mathematical model correlating soil water content, VV and VH backscatter values were developed and tested in this research, as presented in Figure 6, showing the relationship of soil water content (%v) and backscatter values.

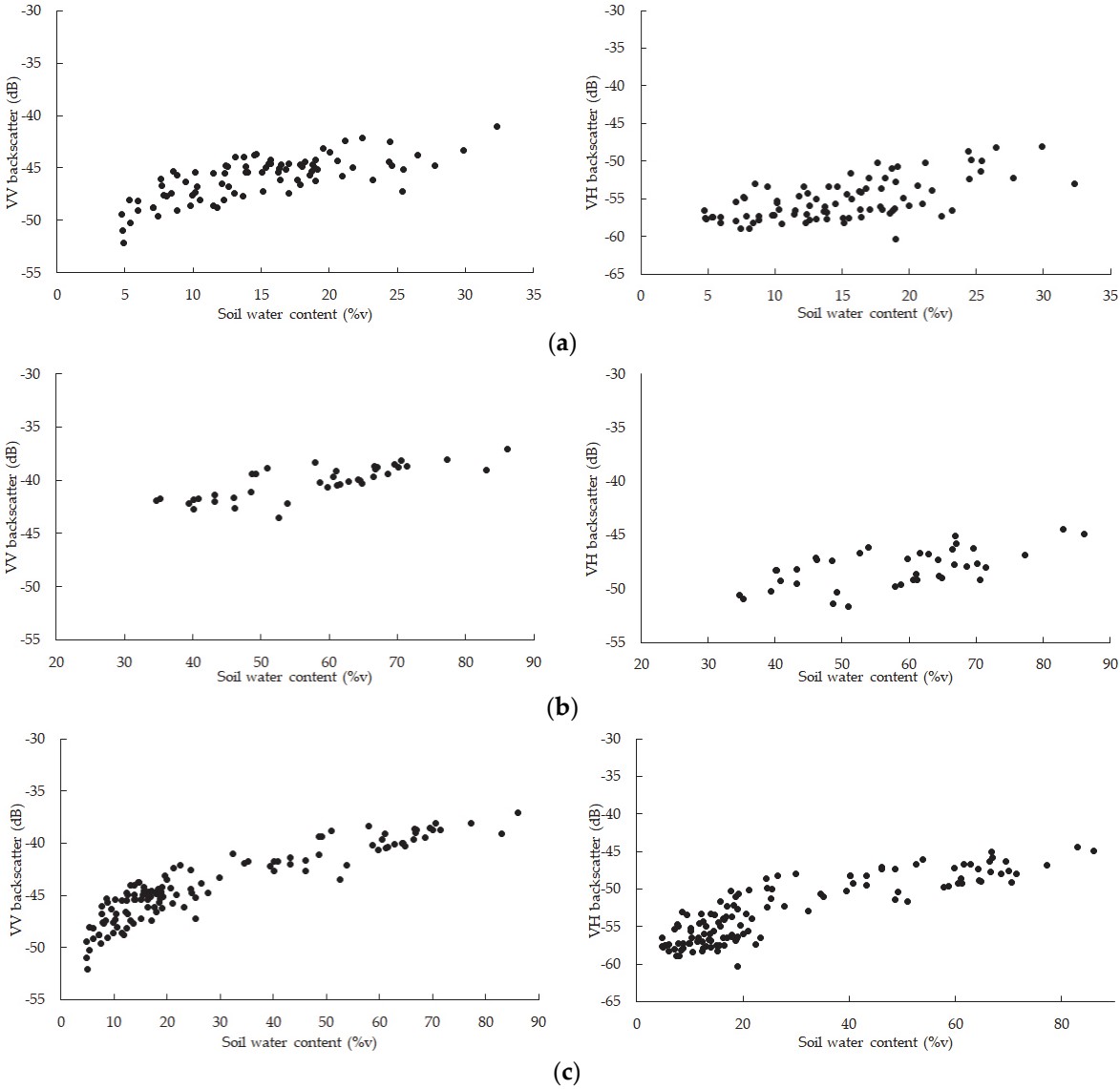

**Figure 6.** Relationships between soil water content (%v) and backscatter values: (**a**) Relationship between VV (**left**) and VH (**right**) backscatter and soil water content in Kediri; (**b**) Relationship between VV (**left**) and VH (**right**) backscatter and soil water content in Sidoarjo; (**c**) The cumulative relationships of VV (**left**) and VH (**right**) backscatter and soil water content.

The multi-polynomial regression method was used to develop a mathematical model to determine the soil water content based on backscatter values, as shown in Equation (4). A total of 80% of the observation data was chosen for model development, and 20% of the data was used for model testing. The relationship between soil water content and backscatter values over unplanted sugar cane plantations in Kediri and Sidoarjo can be defined as:

$$SWC = -2.34 + 210757.27e^{0.2171\sigma_{VV}} + 54709.12e^{0.2003\sigma_{VH}} \tag{4}$$

where $SWC$ is the soil water content (%volume), and $\sigma_{VV}$, and $\sigma_{VH}$ are the backscatter values (dB).

The model has an accuracy and a root-mean-square error (RMSE) of 83.61% and 0.213, respectively. Based on model testing, the results of model accuracy and validation are shown in Table 1 and Figure 7.

**Table 1.** Summary of model accuracy and validation results.

|  | RMSE | MAPE (%) | Accuracy (100 – MAPE) |
|---|---|---|---|
| Model | 0.213 | 16.39 | 83.61 |
| Testing/validation | 0.250 | 18.79 | 81.21 |

Note: A good model is represented with an RMSE ≤ 0.5.

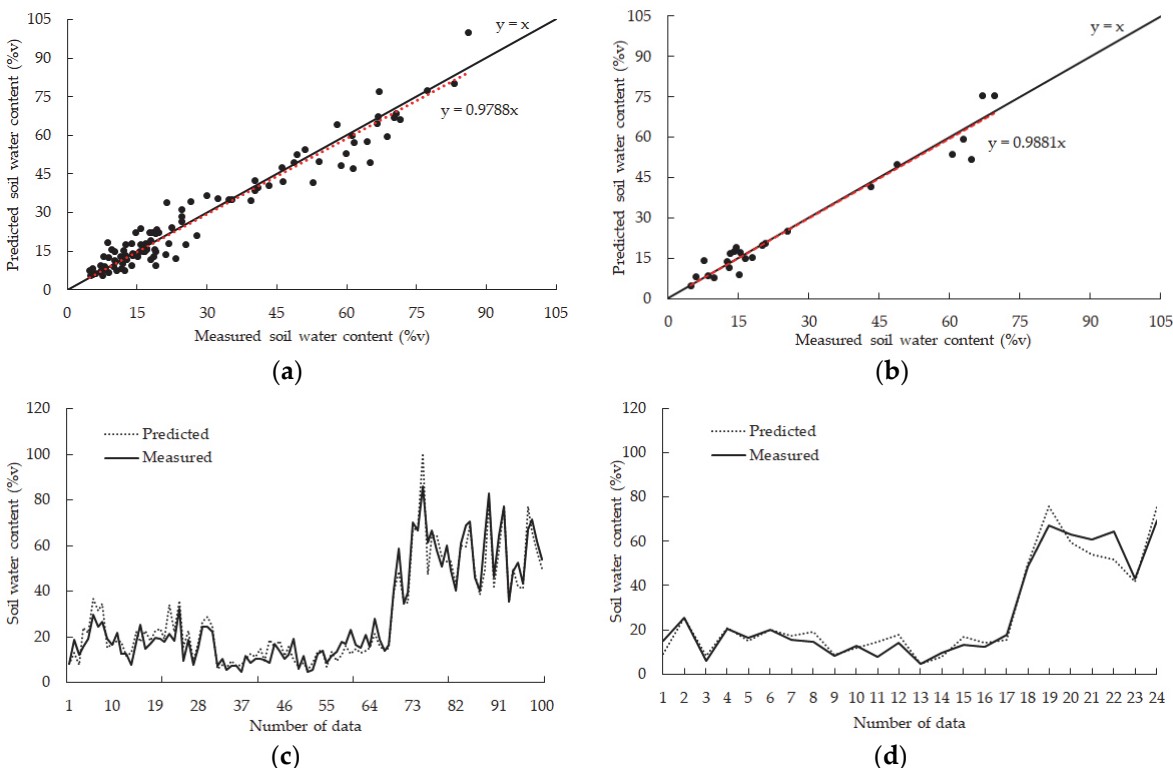

**Figure 7.** Fitting and plotting curves of real and computed values of soil water content: (**a**) Fitting curve of training model; (**b**) Fitting curve of testing model; (**c**) Plotting curve of training model; (**d**) Plotting curve of testing model.

The study proves that Sentinel-1A is a great source and that it is promising for supporting large-scale agricultural management. This finding also conforms with previous studies in exposing the potential of SAR data, such as [65], which proposed a linear relationship between the mean TerraSAR HH backscattering coefficient and the mean volumetric soil water content in the top 0–5 cm of soil for bare agricultural land, as well as [41], which found a linear relationship between ERS1/2 radar signals and volumetric soil water content

gathered in 1995–1997 over the Orgeval watershed. Both models showed good performance with RMSEs of 0.04 and 0.05, respectively.

A spatial distribution map of soil water content was generated from Sentinel-1A imagery based on the empirical model of soil water content provided in Equation (4). The output of the model can be used to calculate soil workability.

To demonstrate the functionality of the model, the spatial distribution of the soil water contents in Kediri on 12 October 2019 and Sidoarjo on 19 October 2019 were calculated and spatially visualized, as shown in Figure 8.

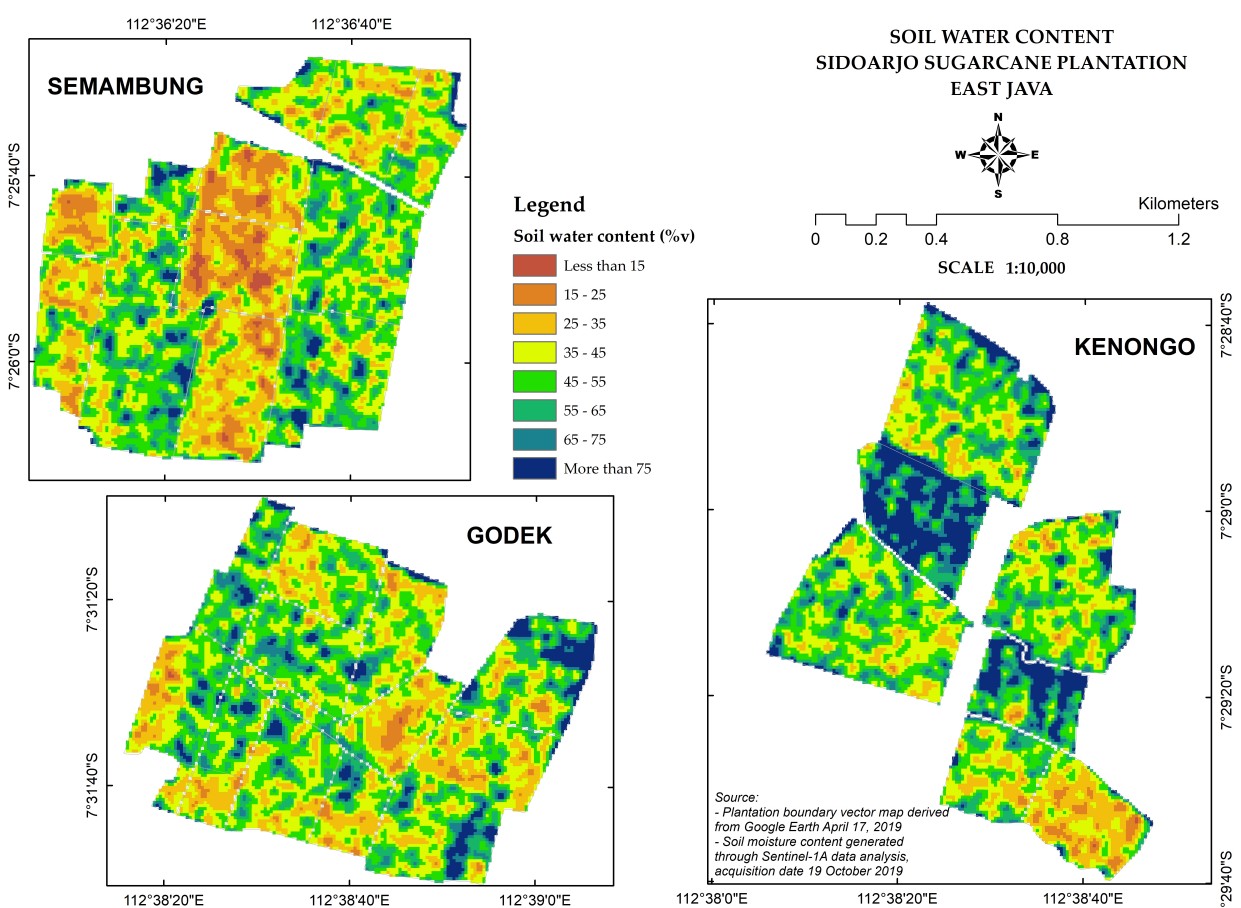

**Figure 8.** Spatial distribution of soil water content in Kediri (**left**) and Sidoarjo (**right**) plantations on 15 October 2019 and 19 October 2019, respectively.

Soil water content at the optimum level of soil workability can be calculated using Equation (3). The liquid limit and plastic limit were obtained through a laboratory analysis. The results of the laboratory analysis showed that the liquid limit and plastic limit of the soil in the Sidoarjo plantation were 58.89% and 24.97%, respectively. The water content at the optimum level of workability and the plasticity index for Sidoarjo plantations were 19.89 %v and 33.91%v, respectively. Thus, the optimum level of soil workability for the Sidoarjo plantation was from 19.89 to 33.91 %v. The dominant sand fraction in the Kediri Plantation caused the soil to be non-plastic; therefore, the plastic limit could not be measured. Referring to [66], suitable soil conditions for tillage can be assessed by knowing the potential matrix (suction) and water retention curve, which describes the relationship between suction and soil water content. The important fixed points on this curve that are relevant to workability are the water content at field capacity (FC), especially at suction between 5 kPa (pF 1.7) and 30 kPa (pF 2.5), and the water content at the inflection point [67]. Based on this, the optimum soil workability level in the Kediri plantation uses the field capacity (28.68 ± 6.22 %v) as the lower limit of the optimum soil workability level. Soil

properties that affect soil workability are soil texture, organic matter content, bulk density and the soil tillage system [68]. The soil workability map can be calculated and presented in the form of a map, as shown in Figure 9.

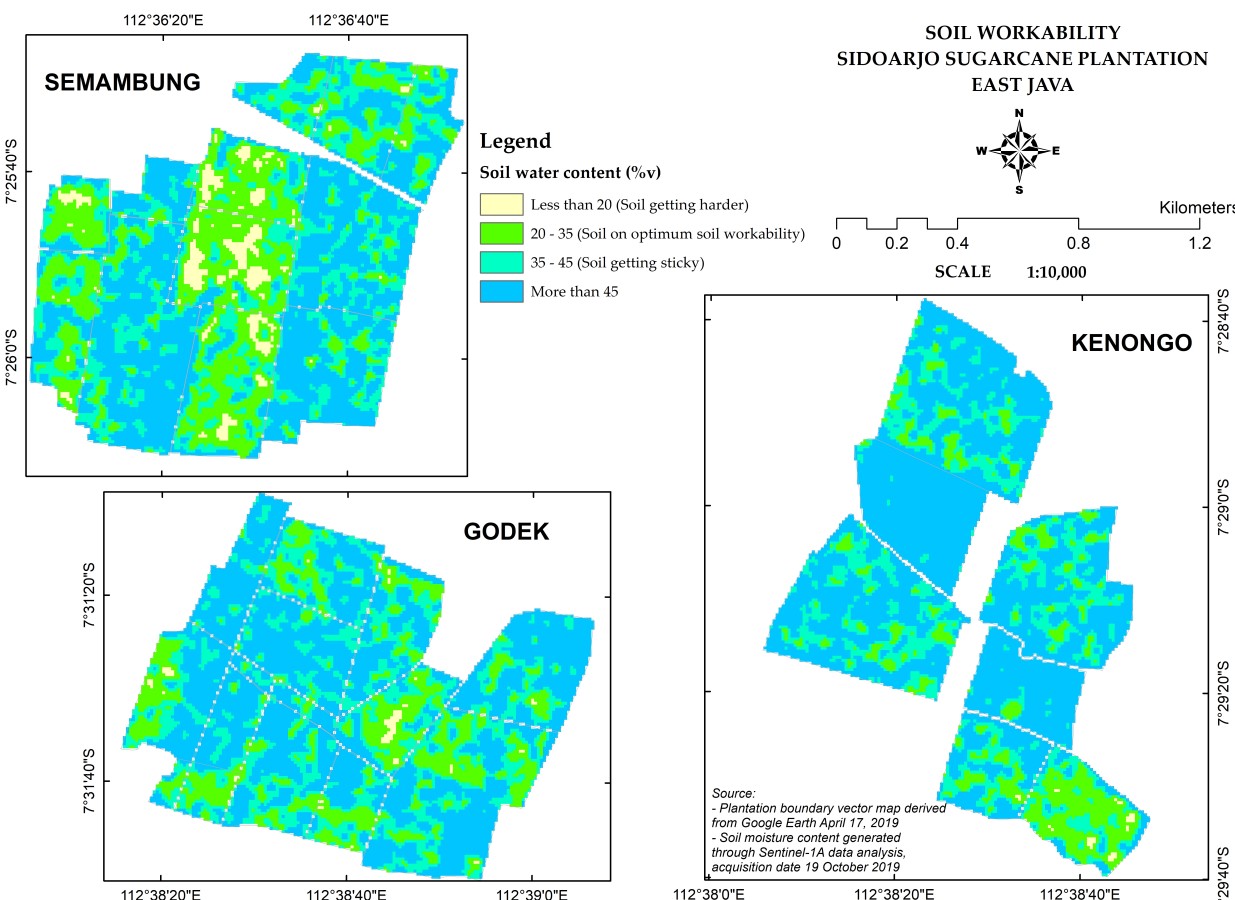

**Figure 9.** Spatial distribution of soil workability in Kediri (**left**) and Sidoarjo (**right**) plantations on 15 October 2019 and 19 October 2019, respectively.

Referring to the soil workability map (Figure 9), sugarcane plantations in Kediri and Sidoarjo which were at the optimum level at the time of image taking were 1633.14 ha and 26.83 ha, respectively. It is shown that the gradation of color corresponds to the level of soil workability. For example, the green color in Figure 9 represents the optimum workability level based on color of the legend of the map. This condition supports optimum tillage activities with minimum negative impact on soil conditions, thus enabling a basis for better route planning and for the scheduling of agricultural machinery operations on a certain field. It is worth pinpointing that knowing the various levels of soil workability can help agricultural management to better handle and manage the spatial variability and diversity of land efficiently.

Although this research is still in its early stages, it has been revealed that the selection of tillage machines for sugarcane companies based on spatial variability is methodologically and technologically possible. Previous related studies have mostly focused on theoretical models of spatial variability in tillage forces and their impact on soils, and this research continues to develop a method and a working model (prototype) for selecting tillage machines based on spatial variability. One of the significant contributions of this research is the method of mapping soil variability (i.e., soil water content and workability) unto suitable tillage machines for specific sugarcane areas.

One of the limitations of this research is the use relatively limited soil samples, and thus it needs to be extended in terms of quantity and sample representation to cover a wide

range of soil conditions in sugarcane plantations in Indonesia. When the proposed method and working model can be applied after some adequate improvements, it can significantly reduce the cost and time for terrain sample work and laboratory analysis. Moreover, using this proposed method, samples can be acquired faster in a much larger area periodically.

## 4. Conclusions

A method to calculate and map soil water content and soil workability based on Sentinel-1A (radar) data was developed and tested. The developed method shows promising performance in terms of root-mean-square error (RMSE), mean absolute percentage error (MAPE) and accuracy. The calculated performance indicators are RMSE = 0.213, MAPE = 16.39% and accuracy = 83.6% for the training model; and RMSE = 0.250, MAPE = 18.79% and accuracy = 81.2% for the testing model.

The spatial distribution of the workability of the soil was calculated and visualized to demonstrate the applicability of the developed method for more comprehensive observation and treatment of site-specific areas. It should be noted that the values of soil water content and soil workability can be used to calculate the specific resistance of the soil, which can then be used to select the most suitable machine for a particular field based on the workability of the soil. Therefore, future work will utilize research products to develop a decision support system for the selection of the optimal machine type for sugarcane field tillage operations.

Due to the limited number and distribution of soil samples, further research is needed to increase the quantity and representativeness of samples to cover more various soil variability in sugarcane plantations in Indonesia.

**Author Contributions:** Conceptualization, H.I.; methodology, K.B.S. and W.H.; validation, W.H.; formal analysis, K.B.S.; investigation, H.I.; resources, W.H., data curation, H.I. and K.B.S.; writing—original draft preparation, H.I.; writing—review and editing, S.K.S.; visualization, H.I., W.H. and S.K.S.; supervision, K.B.S., W.H. and S.K.S.; project administration, H.I.; funding acquisition, H.I. and S.K.S. All authors have read and agreed to the published version of the manuscript.

**Funding:** This research was funded by the Ministry of Education and Culture, Republic of Indonesia, through SEAMEO BIOTROP PhD Grant Program for the fiscal year 2019, grant number 039.1/PSRP/SC/SPK-PNLT/II/2019. The authors thank the Centre for Research and Development of Sugarcane, PTPN X Kediri, Indonesia, for providing field measurement equipment, laboratory facilities and logistical and operational support.

**Institutional Review Board Statement:** Not applicable.

**Informed Consent Statement:** Not applicable.

**Data Availability Statement:** Not applicable.

**Conflicts of Interest:** The authors confirm that there are no known conflicts or competing interests associated with this publication, and there was no significant financial support for this work that could have influenced its outcome.

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
