# Peer review of "A Spatial Distribution Empirical Model of Surface Soil Water Content and Soil Workability on an Unplanted Sugarcane Farm Area Using Sentinel-1A Data towards Precision Agriculture Applications"

_information, doi:10.3390/info13100493_

Round 1

Reviewer 1 Report

Very impressive study and overall very good manuscript. Thanks to the authors.

* Line 31: change "laten heat energy balance" to "surface energy balance"

* Change figures resolution to 300 dpi.

* Support the Discussions with more references and comparisons with previous and similar studies.

Author Response

First of all, we would like to thank and appreciate the corrections and constructive suggestions from the reviewers to improve the quality of the manuscript. Here are our responses to comments and suggestions from reviewers:

Comments and Suggestions for Authors

Very impressive study and overall, very good manuscript. Thanks to the authors.

  • Line 31: change "laten heat energy balance" to "surface energy balance"
  • Response: Suggestions have been followed up accordingly.
  • Change figures resolution to 300 dpi.
  • Response: All pictures/images have been changed into min 300 dpi
  • Support the Discussions with more references and comparisons with previous and similar studies
  • Response: Several relevant references have been added in the introduction and discussion chapters.

Reviewer 2 Report

 Dear authors,

Paper looks interesting, it is a little short, but interesting. I have few questions or comments:

Please give more details related to references in introduction. Paper only poorly expand more details

Please explain, what mean thermal and border noise for radar data.

Please clarify, if 160 terrain measurement  samples is enough for determining regression parameters. What was distance of samples.

Author Response

First of all, we would like to thank and appreciate the corrections and constructive suggestions from the reviewers to improve the quality of the manuscript. Here are our responses to comments and suggestions from reviewers:

Dear authors,

Paper looks interesting, it is a little short, but interesting. I have few questions or comments:

  • Please give more details related to references in introduction. Paper only poorly expands more details

Response:

  • Several relevant references have been added in the introduction and discussion chapters.
  • Please explain, what mean thermal and border noise for radar data.

Response:

  • The explanation of thermal and border noise for radar data have been added to the Materials and Methods section, at line numbers 174 - 184.
  • Please clarify, if 160 terrain measurement samples is enough for determining regression parameters. What was distance of samples.

Response:

  • Although this research is still in its early stages, it is revealed that the selection of tillage machines for sugarcane companies based on spatial variability is methodologically and technologically possible. Previous related studies have mostly focused on theoretical models of spatial variability in tillage forces and their impact on soils, while this research continues to develop a method and a working model (prototype) for selecting tillage machines based on spatial variability. One of the significant contributions of this research is the method of mapping of soil variability (i.e soil water content and workability) unto suitable tillage machines for specific sugarcane areas. One of the limitations of this research is the use relatively limited soil samples, and thus need to be extended in terms of quantity and sample representation to cover a wide range of soil conditions in sugarcane plantations in Indonesia.
  • This statement is also included in the discussion and conclusion chapter.

Reviewer 3 Report

Authors propose to use Sentinel images to analyze soil water content and evaluate soil workability. This new process reduces a lot the current process because nowdays it is requiered to take analyze in laboratory a lot of samples and far away from each other.

I consider it automatizes the analysis process and reduce a lot the costs for companies and farmers.

However, i have some doubts after reading the paper that i consider should be explained for the final version.

In the paper it is said, that volumetric soil water content (%v) and backscatter of SAR data are correlated at the same pixel coordinate positions. But Sentinel images have a resolution of meters for each pixel. How this affect the measures provided in the paper?

Another doubt is about soil workability. Equation 3 provides the optimum soil water content which provides optimum soil workability. It is said that the determines plasticity level. It is clear that comparing the optimal value and the estimated value we have a measure of the workability. But there is not a relation between soil water content and workability. Additionally, workability seems to be related only to two parametes: ??? is the lower plasticity level, and ??? is the upper plasticity level. How we measure this values? Have we have to take a lot of samples in different field locations? Remain the same this values for long time, so we have use these parameters multiple times? Do this values change over time? Can not be estimated with Sentinel images?

Author Response

First of all, we would like to thank and appreciate the corrections and constructive suggestions from the reviewers to improve the quality of the manuscript. Here are our responses to comments and suggestions from reviewers:

Comments and Suggestions for Authors

  • Authors propose to use Sentinel images to analyze soil water content and evaluate soil workability. This new process reduces a lot the current process because nowdays it is requiered to take analyze in laboratory a lot of samples and far away from each other.
  • I consider it automatizes the analysis process and reduce a lot the costs for companies and farmers. However, i have some doubts after reading the paper that i consider should be explained for the final version.
  • In the paper it is said, that volumetric soil water content (%v) and backscatter of SAR data are correlated at the same pixel coordinate positions. But Sentinel images have a resolution of meters for each pixel. How this affect the measures provided in the paper?

Response:

  • Thank you for your comment and suggestion. To relate the soil water content (%v) with backscattering SAR data at the same coordinate position using the point sampling method (with the help of QGIS software). First, the position of the coordinates of the sample points for measuring soil water content is recorded. Second, the coordinates are overlaid with the SAR data, and the point sampling method is applied. Third, as a result, each sample point for measuring soil moisture content will have an equivalent backscatter value.
  • Another doubt is about soil workability. Equation 3 provides the optimum soil water content which provides optimum soil workability. It is said that the determines plasticity level. It is clear that comparing the optimal value and the estimated value we have a measure of the workability. But there is not a relation between soil water content and workability. Additionally, workability seems to be related only to two parametes: ???is the lower plasticity level, and ??? is the upper plasticity level. How we measure this values? Have we have to take a lot of samples in different field locations? Remain the same this values for long time, so we have use these parameters multiple times? Do this values change over time? Can not be estimated with Sentinel images?

Response:

  • An explanation of the relationship between soil moisture content and soil workability has been added to the discussion chapter, on lines 408 to 432.

Round 2

Reviewer 3 Report

It is an interesting paper. It can be a reference to make similar studies at different locations, which could be very useful. Additionally, it describes further research lines to improve the process.